# Chemical Composition of *Ambrosia trifida* L. and Its Allelopathic Influence on Crops

**DOI:** 10.3390/plants10102222

**Published:** 2021-10-19

**Authors:** Jovana Šućur, Bojan Konstantinović, Marina Crnković, Vojislava Bursić, Nataša Samardžić, Đorđe Malenčić, Dejan Prvulović, Milena Popov, Gorica Vuković

**Affiliations:** 1Department of Field and Vegetable Crops, Faculty of Agriculture, University of Novi Sad, Trg Dositeja Obradovića 8, 21000 Novi Sad, Serbia; jovana.sucur@polj.uns.ac.rs (J.Š.); m.crnkovic.95@gmail.com (M.C.); djordje.malencic@polj.uns.ac.rs (Đ.M.); dejanp@polj.uns.ac.rs (D.P.); 2Department of Plant and Environmental Protection, Faculty of Agriculture, University of Novi Sad, Trg Dositeja Obradovića 8, 21000 Novi Sad, Serbia; bojank@polj.uns.ac.rs (B.K.); bursicv@polj.uns.ac.rs (V.B.); milena.popov@polj.uns.ac.rs (M.P.); 3Department of Pesticides and Herbology, Faculty of Agriculture, University of Belgrade, Nemanjina 6, 11080 Belgrade, Serbia; goricavukovic@yahoo.com

**Keywords:** allelopathy, oxidative stress, maize, soybean, sunflower

## Abstract

Phytotoxic substances released by invasive plants have been reported to have anti-pathogen, anti-herbivore, and allelopathic activity. The aim of this study was to determine the allelopathic influence of the *Ambrosia trifida* L. on oxidative stress parameters (the lipid peroxidation process; reduced glutathione (GSH) content; and activity of antioxidant enzymes catalase (CAT), superoxide dismutase (SOD), and peroxidase (PX)) and phenolic compounds (total phenolic and tannin content) in maize (*Zea mays* L.), soybean (*Glycine max* L.), and sunflower (*Helianthus annuus* L.) crops to explore the effect of released allelochemicals through *A. trifida* root on crops. An analysis by HPLC confirmed the presence of protocatechuic acid, p-hydroxybenzoic acid, vanillic acid, and syringic acid as major components in the *A. trifida*. Based on the obtained results for oxidative stress parameters, it can be concluded that the sunflower was the most sensitive species to *A. trifida* allelochemicals among the tested crops. The other two crops tested showed a different sensitivity to *A. trifida*. The soybean did not show sensitivity, while the maize showed sensitivity only 10 days after the sowing.

## 1. Introduction

*Ambrosia trifida* L., commonly named giant ragweed, is a flowering plant belonging to the *Asteraceae* family. The giant ragweed is American (especially North American) in origin and is spreading as a pioneer species worldwide, invading cultivated fields, particularly wheat, corn, and soybean fields. The earliest germination and emergence, as well as the largest seeds and seedlings, give *A. trifida* a decisive advantage over the other plants [1]. Thus, *A. trifida* is listed, along with *Ageratum conyzoides* L. and *Lantana camara* L., as the most economically destructive weed in the world [2]. The same authors concluded that the competition is an important interference mechanism that is responsible for *A. trifida* infestation, but the allelopathy of *A. trifida* would also be an important mechanism.

Allelopathy is defined as “The biochemical interactions among all types of plants, including microorganisms” [3]. One organism produces chemicals that affect another organism and those chemicals are called allelochemicals [4]. Plant phenolic compounds are considered as a major source of allelochemicals [5]. They can be involved in the production of reactive oxygen species (ROS) and can suppress antioxidant enzyme activity-inducing oxidative stress in plants [6,7]. High production of ROS that exceeds the capacity of antioxidant defence enzymes results in oxidative stress and plant cell death [8].

Thus, the phenolic compounds act as antioxidant, antimutagenic and leaf movement-regulating agents, in turn protecting the organism that produces them from the oxidative stress created by the metabolism and their physical environment [9]. Phenolic compounds present in plants are very important constituents that defend plants from infection and injury [10,11]. It has been found that p-hydroxy benzoic acid increases resistance of wheat (*Triticum aestivum* L.) against a pathogen infection, increasing abiotic stress tolerance and impermeability of the cell wall [12]. Furthermore, it was found that syringic acid and p-hydroxy benzoic acid possess antimicrobial activity [13].

Phenolic compounds synthesized by plants are also involved in plant allelopathy inhibiting the growth of the other plant competitors [14,15]. Some weeds inhibit crop growth and development by interfering with them through released allelochemicals. The *Ambrosia* genus biosynthesizes and releases several types of secondary metabolites (phenolics, flavonoids, sesquiterpenes, ambrosin, isabelin, psilostachyin, coronopilin, thiarubrines, and thiophenes) with a broad spectrum of biological activities including allelopathic reactions [2,16]. Specific chemical compounds, i.e., carotane sesquiterpenes, thiarubrines, and thiophenes, were identified from *A. trifida* [2].

Another important non-enzymatic antioxidant molecule is reduced glutathione (GSH), a thiol tripeptide containing γ-glutamate, glycine and, the most important amino acid, cysteine. It has several roles in animal and plant cells and one of them is to buffer the ROS molecules by donating a hydrogen (H) molecule [17]. The donator is a thiol (SH) group from cysteine, which oxidizes to the glutathione disulfide (GSSG) after the donation. This reaction prevents the oxidative stress and the cell death. The ratio of the reduced glutathione to oxidized glutathione (GSH/GSSG) within cells is a measure of the cellular oxidative stress [18].

Interactions between plants and the environment are an important consideration for understanding allelopathy. One of the studies demonstrated that the temperature stress enhances allelochemical inhibition, which indicates that interactions between plants and the environment are important for understanding allelopathy [19].

Phytotoxic substances released by invasive plants have been reported to have anti-pathogen, anti-herbivore, and allelopathic activity [20]. Bearing in mind that *A. trifida* has the ability to spread, becoming an invasive weed, the aim of this study was to determine the allelopathic influence of *A. trifida* on maize (*Zea mays* L.), soybean (*Glycine max* L.), and sunflower (*Helianthus annuus* L.) plants to explore the effect of released allelochemicals through *A. trifida* root on crops. The effect of released allelochemicals of *A. trifida* on oxidative stress parameters (the lipid peroxidation process, reduced glutathione (GSH) content, as well as the activity of antioxidant enzymes (catalase (CAT)), superoxide dismutase (SOD), and peroxidase (PX)) and phenolic compounds (total phenolic and tannin content) in crop plants were examined. The chemical composition of the *A. trifida* was achieved by the HPLC.

*Ambrosia trifida* is locally present in the Central Bačka (Vojvodina) and is expected to spread in the future [21]. The results of the study that dealt with the antioxidant potential of the ragweeds showed that the range of the total phenolic compounds was between 30.0 and 111.1 mg gallic acid equivalents (GAE) per gram of the dry weight of leaves. The contents of all of the measured phenolic compounds in *A. trifida* were pronounced when compared with *A. artemisiifolia* and *Iva xanthifolia*. When it comes to the antioxidant activity, the radical scavenging abilities of the hydroxyl and 1,1-diphenyl-2-picrylhydrazyl (DPPH) were notably higher in the case of *A. trifida* leaves, compared with the other two investigated species [22].

## 2. Results

### 2.1. Phenolic Compounds

The main constituent of the phenolic components (Table 1.) was protocatechuic acid (8.90 ± 0.20 µg/g), followed by p-hydroxybenzoic, vanillic, and syringic acid (4.50 ± 0.01, 3.55 ± 0.01, and 1.75 ± 0.09 µg/g, respectively). The p-Coumaric acid and ferulic acid were represented at the concentration levels of 0.96 ± 0.01 and 0.70 ± 0.01 µg/g, respectively.

### 2.2. The Effects of Ambrosia trifida on Sunflower, Soybean and Maize Total Phenolics and Tannins

The highest amount of the total phenolics was obtained in sunflower leaves at the weed:crop ratio of 3:1, at 10 days after the sowing (30.50 ± 5.18 mg GAE/g DW) (Table 2). There were no significant differences in total tannins in sunflower leaves. Soybean leaves at the weed:crop ratio was 3:1, at 14 days after the sowing had the highest total phenolic content (14.17 ± 0.44 mg GAE/g DW), while the content of tannins was lower in soybean leaves at the weed:crop ratio of 3:1, at 10 days after the sowing (0.41 ± 0.30 mg GAE/g DW), as well as at the weed:crop ratio of 3:1, at 14 days after the sowing (2.69 ± 1.10 mg GAE/g DW). In maize leaves, the highest amount of the total phenolic compounds was obtained at the weed:crop ratio of 3:1, at 10 days after the sowing (22.72 ± 1.49 mg GAE/g DW). The content of the tannins in maize leaves was lower in the treatments compared with the control group 14 days after the sowing.

### 2.3. The Effects of Ambrosia trifida on Sunflower, Soybean and Maize Antioxidant Enzyme Activity, GSH Content and MDA Content

The results of the *A. trifida* effects on the antioxidant enzyme activity, as well as the GSH and MDA content of sunflower, soybean, and maize, are shown in Table 3, Table 4 and Table 5, respectively.

According to Duncan’s multiple range tests, a significant decrease in the activity of the antioxidant enzymes CAT and PX was detected in sunflower leaves 7 days after the sowing at all examined ratios (PX from 20 to 39%, CAT from 43 to 79%). The activity of the SOD was lower only in the weed:crop ratio of 3:1 (28%). Lower production of GSH was detected in the sunflower plants leaves 7 days after the sowing in all examined ratios. After 14 days, there were no significant differences in the GSH production in the weed:crop ratio of 3:1, statistically lower GSH was measured in the weed:crop ratio of 1:3 (19%), while, in the weed:crop ratio of 1:1, significantly higher GSH content was measured (24%). The MDA content, the main end product of the lipid peroxidation process, is used as a biomarker for the oxidative stress. A statistically significant increase in MDA accumulation was recorded in sunflower leaves 14 days after the sowing whereby the weed:crop ratios were 3:1 and 1:1 (123% and 77%, respectively).

A significant decrease in the activity of the SOD was detected in soybean leaves at the weed:crop ratio of 3:1, at 10 and 14 days after the sowing (16% and 55%, respectively). No significant difference was detected in the activity of CAT, while a significant increase in PX activity was detected at 10 days after the sowing at the weed:crop ratio of 3:1 when the highest increase in the activity of the PX was measured (95%). Furthermore, significant production of the GSH was detected in the soybean plants leaves at 7 and 10 days after the sowing at the weed:crop ratio of 3:1 (25% and 28%, respectively), while, in the same ratio at the end of the experimental time, the production of GSH was lower compared to the control (22%). In the soybean leaves, the amount of MDA was lower in the treatments compared with the control during the experimental period (e.g., from 9 to 12%, 14 days after the sowing).

The activity of the SOD showed an increase from 18 to 24% in maize leaves at 10 and 14 days after the sowing at all examined ratios. The highest activity of the CAT was measured at the weed:crop ratio of 1:1, at 7 days after the sowing (119%). The highest activity of PX was detected in maize leaves at the weed:crop ratios of 1:1 and 3:1, at 14 days after the sowing (21% and 34%, respectively). Significant induction of the GSH production was detected in the maize plants leaves 14 days after the sowing in all examined weed:crop ratios (1:3, 1:1, and 3:1) and accumulation of the GSH compared with the control was 16%, 12%, and 20%, respectively. A significant increase in the LP intensity was recorded in maize leaves 10 days after the sowing at the weed:crop ratio of 3:1 (8%). On the other hand, 14 days after the sowing, there was no significant increase in the LP intensity in the treatments compared with the control.

## 3. Discussion

The allelopathic effect of *A. trifida* was investigated in a number of studies. It was reported that *A. trifida* aqueous extracts were significantly inhibitory to the sorghum seedling growth [23]. The effects of the *A. trifida* volatile oil on other plants were examined and it was found that *A. trifida* volatile oil significantly inhibited the germination and growth of maize and wheat but these volatiles also significantly stimulated the germination and growth of barnyard grass weed (*Echinochloa crus-galli*) [24]. Primarily, the volatile allelochemicals of *A. trifida* acting against other plants and stimulating the germination of barnyard grass weed were terpenoids alcanfor, borneol, and borneol acetate.

The main constituent of the phenolic components in *Ambrosia trifida* was protocatechuic acid, followed by p-hydroxy benzoic, vanillic, and syringic acid. Protocatechuic and p-hydroxy benzoic acid formation result from reactions of hydroxylation and methylation that can occur in the aromatic ring of benzoic acid [14].

The effect of *A. trifida* root exudates on wheat (*T. aestivum)* growth was examined and it was found that soil phytotoxicity did not result primarily from *A. trifida* root exudates, but from the residues in the soil [16]. It was also found that 1α-angeloyloxycarotol and 1α-(2-methylbutyroyloxy)-carotol released by *A. trifida* act as allelochemicals and inhibit the growth of wheat. Another study reported that metabolites with carotene skeletons have strong biological activity [25].

The allelochemical stress is a phenomenon when allelochemical compounds suppress the plant growth. The accepted mode of action of many allelochemical compounds is the production of reactive oxygen species (ROS) and induction of oxidative stress. The excessive production of ROS is accompanied by the activation of enzymatic defenses. The activity of the antioxidant enzymes is frequently used as an indicator of the oxidative stress in plants, while an increase in the lipid peroxidation is a widely used stress indicator of the plant membranes [26]. Taking into account that the activity of the antioxidant enzymes in plants can be changed under oxidative stress [26], the changes in the activity of the antioxidant enzymes in sunflower, soybean, and maize leaves could occur as a response to the oxidative stress induced by the allelochemicals released by *A. trifida*. In addition, if allelopathy-provoked stress is strong enough, the activity of the antioxidant enzymes could not prevent the oxidative stress when lipid peroxidation is increased.

The study results showed that the presence of *A. trifida* strongly affected lipid peroxidation in sunflower leaves, particularly at 14 days after the sowing at the weed:crop ratio was 3:1 when the MDA accumulation was the highest, which preceded a significant decrease in the activity of the antioxidant enzymes, i.e., CAT, SOD, and PX (7 days after the sowing), and significantly higher total phenolic content (10 days after sowing). The level of the GSH molecule production differed between the three tested plant species. In sunflower and maize leaves, increased production of GSH was detected at 14 days after the sowing, while, in case of soybean leaves, the production decreased. The differences in the GSH production among plant species were previously reported [27], where the lower production of GSH compared with the control in *Lactuca sativa* cv. Phillipus and a higher production compared with the control in *Brassica oleracea* cv. Bronco were observed. The obtained results showed that the GSH production is in correlation with the MDA production. Specifically, in sunflower and maize leaves, high levels of the GSH and MDA production were detected at 14 days after the sowing, while, in soybean leaves, the levels were low. The reason lies in the fact that the GSH has the role to reduce the occurred lipid peroxides, after which it oxidizes to the GSSG [28].

It was reported that the stress-sensitive plants under the stress conditions accumulate flavonoids, i.e., one group of the plant phenolics which are effective scavengers of ROS and flavonoid productions under stress conditions. This represents negative correlations with an increase in the antioxidant enzyme activity [29]. A significantly higher accumulation of the MDA in sunflower leaves points to the fact that allelopathy-provoked stress by *A. trifida* plants was strong enough and the scavenging effects of the antioxidant enzymes, with decreased activity, could not prevent oxidative burst and induction of the lipid peroxidation.

The maize plants were mildly affected; an increase in LP intensity was recorded at 10 days after the sowing, but at 14 days after the sowing, there was no significant increase in the LP intensity. On the other hand, soybean plants were not affected by the presence of *A. trifida* plants. The amount of MDA was lower in the treatments compared with the control at 14 days after the sowing. The obtained results for the MDA content are in accordance with the results for the activity of antioxidant enzymes when an increase in the activity of antioxidant enzymes (CAT, SOD and PX) was detected. The increased activity of the antioxidant enzymes in soybean and maize plants suggests that the defensive system of the plants prevailed. It was reported that the presence of the neighboring weeds caused an accumulation of the hydrogen peroxide (H_2_O_2_) and the reduction in anthocyanin content in the first leaf of the maize seedlings [30].

Total phenols were produced in higher amounts in all of the examined plants, but the total tannins, which are antioxidants, had lower production compared with the control. This could be correlated with the increased level of the lipid peroxidation in sunflower and maize, since there are bigger amounts of phenolic compounds without the antioxidant role.

In this research, the sunflower was the most sensitive species to the *A. trifida* allelochemicals among the investigated crops. The other two tested crops showed a different sensitivity to *A. trifida*. The soybean did not show any sensitivity, while maize showed sensitivity only at 10 days after the sowing. These findings are in agreement with the results of the study which reported that different crops showed a different sensitivity to the common ragweed (*Ambrosia artemisiifolia* L.) [31]. Among the tested crops (alfalfa, barley, maize, lettuce, tomato, and wheat), the tomato was the most sensitive indicator species to the *A. artemisiifolia* allelopathic residues.

## 4. Materials and Methods

### 4.1. The Validation of the Method

The LC-MD/MS validation parameters were set in accordance with the AOAC guidelines [32]. The validation parameters are shown in Table 6. The validation parameters included retention time (Rt) (expressed in min), precursor product ion, the correlation coefficient (R^2^), repeatability (expressed in %RSD), as well as the limit of quantification (LOQ) (expressed in µg/kg).

### 4.2. Plant Material and Plant Extraction

The giant ragweed was collected in Kosančić village in Serbia (19°28′30.07″ E, 45°30′30.20″ N) in the six-leaf stage in May 2017. The plants were dried at 30 °C for two weeks. Dried leaves were powdered in the mill and stored at +4 °C.

The powdery material (1 g) was soaked in 10 mL of solvent (methanol and HPLC—grade water) and sonicated for 60 min at 55 °C. The supernatant was removed after the centrifugation (4000 rpm for 5 min). The aliquot was evaporated to dryness and reconstituted in 0.5 mL of the mobile phase. After that, the extract was ready for the LC-MS/MS analyses. The method was performed with the modifications compared with the previous studies [33,34].

### 4.3. LC-MS/MS Analysis

The LC was performed with an Agilent 1200 HPLC system equipped with a G1379B degasser, a G1312B binary pump, a G1367D autosampler, and a G1316B column oven. The chromatographic separation was achieved by the Zorbax Eclipse XDB C18 column (150 × 4.6 mm, 1.8 μm) maintained at 30 °C. The mobile phases were 0.1% formic acid in methanol (solvent A) and 0.1% formic acid in Milli-Q water (solvent B). The gradient was 0 min (80% B), 10 min (50% B), 20 min (5% B), 24 min (0% B), 25 min (80% B), with the flow rate of 0.6 mL/min. For the MS analysis, an Agilent 6410 Triple-Quad LC/MS system was applied. Agilent MassHunter data acquisition, qualitative analysis, and quantitative analysis software were used for method development and data acquisition [34]. All of the used solvents were of a chromatography grade and were obtained from J. T. Baker (Deventer, The Netherlands). The gallic acid, ferulic acid, 2-hydroxycinnamic acid, *trans*-cinnamic acid, caffeic acid, p-coumaric acid, chlorogenic acid, quercetin, (+)-catechin, protocatechuic acid, p-hydroxybenzoic acid, vanillic acid, epicatechin, syringic acid, and kaempferol were used as analytical standards. The stock solutions of individual phenol compound were prepared at the concentration of 1.0 mg/mL in methanol. The stock mixture standard (working solution containing all 15 phenol compounds) was obtained by mixing and diluting the stock standards with a mobile phase in the final concentration of 100 μg/mL. The composite mixtures of all phenolic compounds at the appropriate concentrations were used to spike samples in the data validation settings. The acetic acid was of a *p.a*. grade (Carl Roth).

### 4.4. Crop Plant Growth

A competition trial was conducted in vitro at the Laboratory of Biochemistry, Faculty of Agriculture, Novi Sad, Serbia. The crop (maize (*Zea mays* L.), sunflower (*Helianthus annuus* L.), soybean (*Glycine max* (L.) Merr.), and the weed (*Ambrosia trifida* L.) seeds were surface-sterilized with 3% H_2_O_2_ (*v*/*v*) and washed with deionised water. The seeds of crops and *A. trifida* were sown in pots 40 × 30 cm, at a three-term ratio of 1:3, 1:1, and 3:1 (weed:crop), for each crop separately, while the control pots contained crops only. The sowing substrate consisted of garden humus and sterilized sand at a ratio 2:1. After the sowing, the pots were covered with aluminum foil and put in a growing chamber under the controlled conditions (28 °C, 60% relative humidity, a photoperiod of 18 h, and a light intensity of 10.000 lx). The irrigation was carried out with the sterilized water every day. The evaluation was conducted at 7, 10, and 14 days after the emergence.

### 4.5. Determination of Total Phenolics and Tannins

The air-dried crop leaves (from each growth condition), collected at 7, 10, and 14 days after the emergence, were extracted with 10 mL of 70% ethanol. After 24 h, the extracts were filtered through Whatman No. 4 filter paper and stored at 4 °C until analyzed.

The total phenolics (TP) and tannins (TT) were determined according to the Folin–Ciocalteu method [35]. The leaf extracts were mixed with deionized water, 20% sodium carbonate, and Folin–Ciocalteu reagent diluted with distilled water in proportion 1:2. The absorbance of the reaction mixture was measured after incubation at the room temperature for 30 min at 720 nm using an UV/VIS spectrophotometer (Thermo Scientific Evolution 220, Waltham, MA, USA). The standard curve for TP and TT contents was plotted using gallic acid (GA) solution (0.1–2.0 mg/mL). The data were expressed as mg of gallic acid equivalent per gram of dry weight (mg GAE/g DW).

### 4.6. Determination of the Oxidative Stress Parameters

For the determination of the oxidative stress parameters, i.e., activity of antioxidant enzyme activity, content of reduced glutathione (GSH), and intensity of lipid peroxidation, 2 g of fresh crop plant material (leaves from each growth condition: control and three ratios 1:3, 1:1, 3:1 (weed:crop)), collected at 7, 10, and 14 days after the emergence, were crushed and homogenized in 10 mL of phosphate buffer (0.1 M, pH 7.0) and prepared in-house. After centrifugation, clear supernatants were used for further biochemical analyses. Biochemical assays were carried out spectrophotometrically using an UV/VIS spectrophotometer (Thermo Scientific Evolution 220 (USA)).

The catalase (CAT) (EC 1.11.1.6) activity was determined according to the following method [36]. The decomposition of the H_2_O_2_ was followed by a decrease in absorbance at 240 nm. The enzyme extract was added to the assay mixture containing 50 mM of potassium phosphate buffer (pH 7.0) prepared in-house and 10 mM of H_2_O_2_. The activity of the enzyme was expressed as U per gram of fresh weight (U/g FW).

The assay of the superoxide dismutase (SOD) (EC 1.15.1.1) activity [37] is based on the ability of the enzyme extracts to inhibit the photochemical reduction in the nitro blue tetrazolium (NBT) chloride. The reaction medium was prepared by mixing 50 mM of phosphate buffer (pH 7.8), 75 μM of NBT, 13 mM of L-methionine, 0.1 mM of EDTA, 2 μM of riboflavin, and enzyme extract. It was kept under a fluorescent lamp for 30 min, and the absorbance was read at 560 nm. One unit of the SOD activity was defined as the amount of enzymes required to inhibit reduction in NBT by 50%. The activity of the enzyme was expressed as U per gram of fresh weight (U/g FW).

The peroxidase (PX) (EC 1.11.1.7) activity was measured using pyrogallol as a substrate. This method [38] is based on the purpurogallin content measurement, i.e., a product of the pyrogallol oxidation. The enzyme extract was added to the assay mixture containing 180 mM of pyrogallol and 2 mM of H_2_O_2_. The absorbance was recorded at 430 nm. The activity of the enzyme was expressed as U per gram of fresh weight (U/g FW).

The reduced glutathione (GSH) was determined [39] and expressed as μmol GSH per gram of fresh weight (µmol GSH/g FW).

The intensity of the lipid peroxidation (LP) was determined using the thiobarbituric acid (TBA) test at 532 nm [34]. The enzyme extract was incubated with 20% trichloroacetic acid (TCA) containing 0.5% thiobarbituric acid for 40 min at 95 °C. The reaction was stopped by cooling on ice for 10 min, after which the product was centrifuged at 10,000 rpm for 15 min. The total amount of TBA-reactive substances was given as nmol of malondialdehyde (MDA) equivalents per gram of fresh weight (nmol MDA/g FW).

### 4.7. Statistical Analysis

All measurements were performed in triplicates. The values of the biochemical parameters were expressed as a mean ± standard error of mean and tested by ANOVA, followed by a comparison of the means by Duncan’s multiple range test (*p* < 0.05). The data were analyzed using STATISTICA for Windows, version 11.0. Comparable percentage was calculated with the formula:Δ (%) = (100 × sample/control) − 100

## 5. Conclusions

According to the obtained results for the oxidative stress parameters, it can be concluded that the tested crops showed different sensitivity to *A. trifida*. The highest amount of MDA was detected in sunflower leaves, particularly 14 days after the sowing at the weed:crop ratio of 3:1, thus the sunflower was the most sensitive crop tested. Maize showed mild sensitivity, while the soybean did not show sensitivity in these conditions.

## Figures and Tables

**Table 1 plants-10-02222-t001:** The identified and quantified phenolic compounds in the *Ambrosia trifida*.

Phenolic Compounds	mean ± SD µg/g *
gallic acid	nd
ferulic acid	0.70 ± 0.01
2-hydroxycinnamic acid	nd
*trans*-cinnamic acid	nd
caffeic acid	nd
p-coumaric acid	0.96 ± 0.00
chlorogenic acid	nd
quercetin	nd
(+)-catechin	nd
protocatechuic acid	8.90 ± 0.28
p-hydroxybenzoic acid	4.50 ± 0.00
vanillic acid	3.55 ± 0.00
epicatechin	nd
syringic acid	1.75 ± 0.09
kaempferol	nd

* Data represent the means of the three replicates ± standard deviation; nd = not defined.

**Table 2 plants-10-02222-t002:** Total phenolics (TP) and total tannins (TT) in leaves of sunflower, soybean and maize plants of the three-term ratio 1:3, 1:1, 3:1 (weed-crop).

Time	Ratio	Sunflower		Soybean		Maize	
		TP	TT	TP	TT	TP	TT
7 days	C	11.67 ± 0.90 ^a^	4.36 ± 0.46 ^ab^	9.93 ± 1.18 ^a^	3.09 ± 1.12 ^b^	14.82 ± 2.13 ^ac^	5.65 ± 1.99 ^bc^
	1:3	7.95 ± 0.92 ^a^	3.35 ± 0.34 ^ab^	11.23 ± 0.89 ^a^	7.03 ± 0.67 ^d^	14.34 ± 1.74 ^ac^	8.00 ± 1.67 ^c^
	1:1	11.74 ± 0.89 ^a^	4.76 ± 0.07 ^ab^	10.12 ± 0.64 ^a^	5.90 ± 0.60 ^cd^	15.52 ± 1.20 ^ac^	6.77 ± 0.82 ^c^
	3:1	7.18 ± 1.14 ^a^	3.23 ± 0.17 ^ab^	10.59 ± 1.09 ^a^	5.19 ± 1.11 ^bcd^	13.75 ± 0.96 ^b^	5.86 ± 1.00 ^bc^
10 days	C	11.20 ± 1.02 ^a^	3.70 ± 0.22 ^ab^	10.23 ± 0.78 ^a^	4.30 ± 0.53 ^bc^	15.86 ± 0.48 ^ac^	7.98 ± 0.22 ^c^
	1:3	6.83 ± 0.78 ^a^	2.89 ± 0.15 ^a^	11.08 ± 0.47 ^a^	4.93 ± 0.77 ^bcd^	17.28 ± 0.57 ^c^	8.40 ± 0.42 ^c^
	1:1	9.09 ± 1.34 ^a^	3.88 ± 0.26 ^ab^	10.90 ± 0.45 ^a^	4.33 ± 1.01 ^bc^	14.76 ± 1.09 ^ac^	5.52 ± 0.92 ^bc^
	3:1	30.50 ± 5.18 ^b^	5.01 ± 0.17 ^b^	11.05 ± 0.94 ^a^	0.41 ± 0.30 ^a^	22.72 ± 1.49 ^a^	5.70 ± 0.61 ^bc^
14 days	C	12.81 ± 0.55 ^a^	5.04 ± 0.38 ^b^	10.32 ± 0.51 ^a^	6.10 ± 0.31 ^c^	16.55 ± 0.12 ^c^	12.85 ± 0.22 ^a^
	1:3	14.26 ± 0.51 ^a^	4.42 ± 0.44 ^ab^	10.60 ± 0.35 ^a^	4.22 ± 0.18 ^bc^	15.05 ± 0.18 ^ac^	6.91 ± 0.16 ^c^
	1:1	13.67 ± 1.19 ^a^	5.00 ± 0.51 ^b^	10.39 ± 0.53 ^a^	3.66 ± 0.47 ^bc^	14.64 ± 0.73 ^ac^	5.70 ± 0.76 ^bc^
	3:1	16.14 ± 1.35 ^a^	5.10 ± 0.14 ^b^	14.17 ± 0.44 ^b^	2.69 ± 1.10 ^b^	12.72 ± 0.80 ^b^	3.12 ± 0.17 ^b^

TP (mgGAE/gDW); TT (mgGAE/gDW). The data are mean values ± standard error. ^a–d^ Values without the same superscripts within each column differ significantly (*p* < 0.05).

**Table 3 plants-10-02222-t003:** The activity of the antioxidant enzymes, reduced glutathione (GSH) content, and malondialdehyde (MDA) content in leaves of sunflower plants of the three-term ratio 1:3, 1:1, 3:1 (weed-crop).

Time	Ratio	CAT	SOD	PX	GSH	MDA
7 days	C	0.29 ± 0.16 ^a^	1241.71 ± 1150.06 ^d^	0.77 ± 0.65 ^c^	0.20 ± 0.20 ^a^	35.04 ± 13.26 ^a^
	1:3	0.06 ± 0.07 ^b^	1428.85 ± 1337.20 ^abc^	0.47 ± 0.35 ^a^	0.16 ± 0.15 ^b^	32.16 ± 10.38 ^a^
	1:1	0.17 ± 0.04 ^c^	1467.84 ± 1376.18 ^a^	0.55 ± 0.43 ^ab^	0.16 ± 0.16 ^b^	25.21 ± 3.43 ^a^
	3:1	0.09 ± 0.04 ^c^	888.89 ± 797.24 ^e^	0.62 ± 0.50 ^abc^	0.17 ± 0.16 ^b^	35.26 ± 13.48 ^a^
10 days	C	0.10 ± 0.03 ^a^	1396.19 ± 1304.54 ^abc^	0.63 ± 0.51 ^abc^	0.180.17 ^a^	42.73 ± 20.96 ^ab^
	1:3	0.12 ± 0.01 ^a^	1443.81 ± 1352.16 ^ab^	0.61 ± 0.49 ^abc^	0.20 ± 0.19 ^b^	34.83 ± 13.05 ^a^
	1:1	0.07 ± 0.06 ^a^	1304.76 ± 1213.11 ^c^	0.62 ± 0.50 ^abc^	0.17 ± 0.16 ^c^	55.66 ± 33.88 ^ab^
	3:1	0.15 ± 0.02 ^a^	1329.52 ± 1237.87 ^b^	0.70 ± 0.58 ^b^	0.17 ± 0.16 ^a^	45.62 ± 23.84 ^ab^
14 days	C	0.11 ± 0.02 ^a^	1505.24 ± 1413.59 ^a^	0.81 ± 0.69 ^d^	0.21 ± 0.20 ^a^	40.49 ± 18.71 ^ab^
	1:3	0.09 ± 0.04 ^a^	1456.86 ± 1365.21 ^ab^	0.57 ± 0.49 ^ab^	0.17 ± 0.16 ^b^	39.53 ± 17.75 ^ab^
	1:1	0.17 ± 0.04 ^a^	1472.55 ± 1380.90 ^a^	0.89 ± 0.77 ^e^	0.26 ± 0.25 ^c^	71.69 ± 49.91 ^b^
	3:1	0.12 ± 0.01 ^a^	1488.23 ± 1396.58 ^a^	0.91 ± 0.79 ^e^	0.20 ± 0.19 ^a^	90.38 ± 68.61 ^c^

Activity of antioxidant enzymes (U/g FW); CAT (catalase, U/g FW); SOD (superoxide dismutase, U/g FW); PX (peroxidase, U/g FW); GSH content (reduced glutathione, µmol GSH/g FW); MDA content (nmol MDA/g FW). The data are mean values ± standard error. ^a–e^ Values without the same superscripts within each column differ significantly (*p* < 0.05).

**Table 4 plants-10-02222-t004:** The activity of the antioxidant enzymes, reduced glutathione (GSH) and malondialdehyde (MDA) content in leaves of soybean plants of the three-term ratio 1:3, 1:1, and 3:1 (weed-crop).

Time	Ratio	CAT	SOD	PX	GSH	MDA
7 days	C	0.13 ± 0.03 ^a^	1062.38 ± 956.17 ^a^	1.50 ± 1.20 ^a^	0.25 ± 0.24 ^a^	91.13 ± 84.96 ^a^
	1:3	0.06 ± 0.04 ^a^	877.19 ± 770.99 ^c^	0.83 ± 0.53 ^b^	0.20 ± 0.19 ^b^	80.34 ± 74.17 ^b^
	1:1	0.13 ± 0.03 ^a^	1027.29 ± 921.09 ^ac^	1.05 ± 0.76 ^ab^	0.20 ± 0.19 ^b^	55.66 ± 49.49 ^f^
	3:1	0.08 ± 0.02 ^a^	1109.16 ± 1002.96 ^a^	1.27 ± 0.97 ^ab^	0.31 ± 0.30 ^c^	74.68 ± 68.51 ^be^
10 days	C	0.07 ± 0.03 ^a^	1308.57 ± 1202.37 ^b^	1.20 ± 0.91 ^ab^	0.24 ± 0.23 ^a^	109.19 ± 103.01 ^cd^
	1:3	0.09 ± 0.01 ^a^	1369.52 ± 1263.32 ^b^	1.05 ± 0.76 ^ab^	0.20 ± 0.19 ^b^	101.39 ± 95.22 ^c^
	1:1	0.09 ± 0.01 ^a^	1022.86 ± 916.65 ^ac^	1.41 ± 1.12 ^a^	0.26 ± 0.25 ^c^	67.63 ± 61.45 ^e^
	3:1	0.20 ± 0.10 ^a^	1099.05 ± 992.84 ^a^	2.34 ± 2.05 ^cd^	0.31 ± 0.30 ^d^	85.36 ± 79.19 ^ab^
14 days	C	0.08 ± 0.02 ^a^	1392.16 ± 1285.95 ^b^	2.39 ± 2.09 ^cd^	0.31 ± 0.30 ^a^	114.10 ± 107.93 ^c^
	1:3	0.11 ± 0.01 ^a^	1447.06 ± 1340.86 ^b^	2.01 ± 1.72 ^c^	0.32 ± 0.31 ^a^	100.32 ± 94.15 ^d^
	1:1	0.15 ± 0.05 ^a^	1433.33 ± 1327.13 ^b^	2.51 ± 2.21 ^d^	0.29 ± 0.28 ^b^	103.74 ± 97.57 ^d^
	3:1	0.11 ± 0.01 ^a^	623.53 ± 517.33 ^d^	2.36 ± 2.06 ^cd^	0.24 ± 0.23 ^c^	100.21 ± 94.04 ^d^

Activity of antioxidant enzymes (U/g FW); CAT (catalase, U/g FW); SOD (superoxide dismutase, U/g FW); PX (peroxidase, U/g FW); GSH content (reduced glutathione, µmol GSH/g FW); MDA content (nmol MDA/g FW). The data are mean values ± standard error. ^a–e^ Values without the same superscripts within each column differ significantly (*p* < 0.05).

**Table 5 plants-10-02222-t005:** The activity of the antioxidant enzymes, reduced glutathione (GSH) and malondialdehyde (MDA) content in leaves of maize plants of the three-term ratio 1:3, 1:1, 3:1 (weed-crop).

Time	Ratio	CAT	SOD	PX	GSH	MDA
7 days	C	0.33 ± 0.16 ^a^	1335.28 ± 1258.50 ^e^	1.12 ± 0.96 ^c^	0.20 ± 0.19 ^a^	49.25 ± 33.69 ^a^
	1:3	0.23 ± 0.07 ^a^	1085.77 ± 1008.99 ^b^	1.00 ± 0.83 ^d^	0.22 ± 0.20 ^a^	43.80 ± 28.24 ^a^
	1:1	0.72 ± 0.55 ^b^	1042.88 ± 966.10 ^b^	1.32 ± 1.16 ^abc^	0.20 ± 0.19 ^a^	39.53 ± 23.96 ^a^
	3:1	0.20 ± 0.04 ^a^	807.02 ± 730.23 ^d^	1.04 ± 0.88 ^d^	0.21 ± 0.20 ^a^	51.17 ± 35.61 ^a^
10 days	C	0.22 ± 0.06 ^a^	754.29 ± 677.50 ^d^	1.43 ± 1.27 ^ab^	0.23 ± 0.22 ^a^	50.43 ± 34.86 ^a^
	1:3	0.22 ± 0.06 ^a^	923.81 ± 847.02 ^a^	1.21 ± 1.05 ^a^	0.25 ± 0.24 ^b^	44.23 ± 28.67 ^a^
	1:1	0.34 ± 0.18 ^a^	910.48 ± 833.69 ^ac^	1.53 ± 1.37 ^b^	0.19 ± 0.17 ^c^	48.50 ± 32.94 ^a^
	3:1	0.21 ± 0.04 ^a^	935.24 ± 858.45 ^abc^	1.42 ± 1.26 ^ab^	0.33 ± 0.32 ^d^	54.06 ± 38.49 ^a^
14 days	C	0.14 ± 0.02 ^a^	851.15 ± 774.37 ^c^	1.33 ± 1.17 ^abc^	0.25 ± 0.24 ^a^	58.01 ± 42.45 ^a^
	1:3	0.12 ± 0.04 ^a^	1012.58 ± 935.79 ^ab^	1.35 ± 1.19 ^abc^	0.29 ± 0.27 ^b^	58.65 ± 43.09 ^a^
	1:1	0.31 ± 0.15 ^a^	1018.87 ± 942.08 ^ab^	1.61 ± 1.45 ^e^	0.28 ± 0.27 ^b^	58.01 ± 42.45 ^a^
	3:1	0.23 ± 0.07 ^a^	1008.39 ± 931.60 ^ab^	1.78 ± 1.62 ^f^	0.30 ± 0.29 ^b^	52.88 ± 37.32 ^a^

Activity of antioxidant enzymes (U/g FW); CAT (catalase, U/g FW); SOD (superoxide dismutase, U/g FW); PX (peroxidase, U/g FW); GSH content (reduced glutathione, µmol GSH/g FW); MDA content (nmol MDA/g FW). The data are mean values ± standard error. ^a–e^ Values without the same superscripts within each column differ significantly (*p* < 0.05).

**Table 6 plants-10-02222-t006:** The LC-MS/MS validation parameters.

Phenolic Compounds	Rt (min)	Precursor Product Ion	R^2^	Repeatability (RSD, %)	LOQ (µg/kg)
gallic acid	4.12	169→125	0.9967	9.5	0.1
ferulic acid	12.62	193→134 193→177.5	0.9988	12.1	0.1
2-hydroxycinnamic acid	7.25	163→117 163→119	0.9954	5.9	0.1
*trans*-cinnamic acid	13.27	147→147	0.9816	8.1	0.1
caffeic acid	10.86	179→135	0.9995	11.7	0.1
p-coumaric acid	12.20	163→93 163→119	0.9990	8.6	0.1
chlorogenic acid	10.05	353→191	0.9990	9.2	0.1
quercetin	15.13	301→151 301→179	0.9969	7.4	0.1
(+)-catechin	9.79	289→205 289→245	0.9980	4.3	0.1
protocatechuic acid	7.53	153→109	0.9995	4.3	0.1
p-hydroxybenzoic acid	9.62	137→93	0.9980	9.2	0.1
vanillic acid	10.68	167→108	0.9996	11.7	0.1
epicatechin	11.14	189→245	0.9986	5.9	0.1
syringic acid	11.17	197→182	0.9998	4.3	0.1
kaempferol	15.65	285→169 285→285	0.9968	4.2	0.1

## Data Availability

The data is contained within this manuscript. All data, tables and figures in this manuscript are original.

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
