# Peer review of "Chemical Composition of Ambrosia trifida L. and Its Allelopathic Influence on Crops"

_plants, 2021, doi:10.3390/plants10102222_

Round 1

Reviewer 1 Report

Dear authors, please find my comments in pdf attached

Author Response

Response to Reviewer 1 Comments

Response: Dear Reviewer 1, we would like to thank you for your comments which have greatly contributed to the quality of our paper. It was rather difficult to correct the entire paper, since there were 3 Reviewers who asked for completely different corrections. We have done our best to accept all your suggestions and you can find the corrections in the attached paper highlighted in blue. We sincerely hope that you will find our effort and the work we have put in correcting the paper useful and that our paper will now meet your criteria for being published in your journal.

Point 1: Put the reference [2] here. And then you can say that same authors concluded that competition...
Response: Corrected and highlighted in blue.
Point 2: Again, you need a reference. Anyway, I suggest using only one of these definitions of allelopathy, rather than 3 that you used.
Response: The reference was added and only one of the definitions was used. The corrections are highlighted in blue.
Point 3: reference!
Response: The reference was added and highlighted in blue.
Point 4: The sentence is not finished, maybe you can connect this sentence with the next.
Response: Corrected and highlighted in blue.
Point 5: I suggest to add in Introduction the importance of A. trifida as invasive weed species in Serbia, the distribution as well as crops or ruderal places were usally A. trifida appears. Also, is there any literature data about phenolic compounds from A. trifiida or other Ambrosia species on oxidative stress parameters, please mentioned it otherwise mention that specific literature data could not be found.
Response: We have added the part about the importance of the A. trifida as an invasive weed in Serbia, as well as the literature data about phenolic compounds. The changes are highlighted in blue.
Point 6: You must consult the Intruction for authors, in this Journal Material and methods is written after Discussion. This means that you also need to reorder the reference numbers throught the Ms.
Response: We have changed the order of the chapters according to the Instructions for authors, as well as the order of the references throughout the paper.
Point 7: Can you name the qrowth stage of A. trifida at the time of collection
Response: We have added the growth stage of A. trifida and highlighted the changes in blue.
Point 8: when? days after sowing or BBCH
Response: We have added the growth stage of the crop and highlighted the changes in blue.
Point 9: same as previous, at which crop growth stage?
Response: We have added the growth stage of the crop and highlighted the changes in blue.
Point 10: This is a part for M M so eventually you can add this table in M M or even better in Supplement material or Appendix. In results you report only phenolic compounds found in A. Trifida                                                               Response: We have moved the part of the table considering validation parameters to the Materials and Methods part of the paper. The changes are highlighted in red.
Point 11: (Table 1)
Response: Unfortunately, we couldn't fulfil this request since the Academic Editor requested that we put the part of the table considering validation parameters in the “Materials and Methods” and not in the Discussion part where you marked the spot where the table should be.
Point 12: This part belongs rather in the introductory chapter, as the subject here is weed-plant allelopathy. I propose to include it again in the introduction
Response: We have moved that part to the Introduction and highlighted the changes in blue.
Point 13: It has been found that .... (17). Furhtermore, syringic acid and p..... (18)
Response:We have made the corrections and highlighted them in blue.
Point 14: this also rather belongs to introduction
Response: We have moved that part to the Introduction and highlighted the changes in blue.
Point 15: Interactions between plants and the environment are an important consideration for 256 understanding allelopathy. Einhellig and Eckrick [22] demonstrated that temperature 257 stress enhances allelochemical inhibition which indicates that interactions between plants 258 and the environment are important for understanding allelopathy.
Response: We have moved that part to the Introduction and highlighted the changes in blue.
Point 16: omit this, mentioned before
Response:We have deleted that sentence.
Point 17: Maybe you can remove this paragraf at the beginig of discussion
Response: We have moved that part to the beginning of the Discussion and highlighted it in blue.

Reviewer 2 Report

Dear Authors,

In my opinion, the article is correctly written. The only thing you should pay attention to is the order of subsections that are not presented according to the manuscript template. After the correction, please remember to change the numbering of the cited articles. Besides, I have no major comments to this paper. They fully recommend it for publication. 

Author Response

Response to Reviewer 2 Comments

Response: Dear Reviewer 2, we would like to thank you for your comments which have greatly contributed to the quality of our paper. It was rather difficult to correct the entire paper, since there were 3 Reviewers who asked for completely different corrections. We have done our best to accept all your suggestions and you can find the corrections in the attached paper highlighted in pink. We sincerely hope that you will find our effort and the work we have put in correcting the paper useful and that our paper will now meet your criteria for being published in your journal.

Point 1: delete please it is in the title
Response:We have corrected that.
Point 2: full name, please
Response: Corrected and highlighted in pink.
Point 3: good, please add the same in other tables abbrreviations with full name
Response: Corrected and highlighted in pink.
Point 4: full name
Response: Corrected and highlighted in pink.
Point 5: Is it correct? full name
Response: Corrected and highlighted in pink.
Point 6: full name add below
Response: Corrected and highlighted in pink.
Point 7: what does it mean LOP? Explain it.
Response: Corrected and highlighted in pink.
Point 8: the same question as like in table 3
Response: Corrected and highlighted in pink.
Point 9: dots please
Response: Corrected and highlighted in pink.
Point 10: please look above
Response: Corrected and highlighted in pink.
Point 11: I suggest full name in the title
Response: Corrected and highlighted in pink.
Point 12: dots please
Response: Corrected and highlighted in pink.
Point 13: first time add full name
Response: Corrected and highlighted in pink.
Point 14: small letters
Response: Corrected and highlighted in pink.
Point 15: Journals abbrreviations should be with dots
Response: Corrected and highlighted in pink.
Point 16: Weed Biol Manag
Response: Corrected and highlighted in pink.
Point 17: J Nov Appl Sci
Response: Corrected and highlighted in pink.
Point 18: bold
Response: Corrected and highlighted in pink.
Point 19: Acta Physiol Plant
Response: Corrected and highlighted in pink.
Point 20: Stress in Plants
Response: Corrected and highlighted in pink.
Point 21: Physiol Mol Plant Path
Response: Corrected and highlighted in pink.
Point 22: J Plant Physiol
Response: Corrected and highlighted in pink.
Point 23: Chem
Response: Corrected and highlighted in pink.
Point 24: Extracts
Response: Corrected and highlighted in pink.
Point 25: Int J Mol Sci
Response: Corrected and highlighted in pink.
Point 26: Biol Planta
Response: Corrected and highlighted in pink.
Point 27: small
Response: Corrected and highlighted in pink.
Point 28: J Agric Sci
Response: Corrected and highlighted in pink.
Point 29: Agric Ecosyst Environ
Response: Corrected and highlighted in pink.
Point 30: J Nat Prod
Response: Corrected and highlighted in pink.
Point 31: J Chem Ecol
Response: Corrected and highlighted in pink.
Point 32: long pause
Response: Corrected and highlighted in pink.
Point 33: Plant Sci
Response: Corrected and highlighted in pink.
Point 34: Weed Sci
Response: Corrected and highlighted in pink.
Point 35: J Chem Ecol
Response: Corrected and highlighted in pink.
Point 36: J
Response: Corrected and highlighted in pink.
Point 37: Crop Prot
Response: Corrected and highlighted in pink.

Reviewer 3 Report

Dear authors

The manuscript deals with a relevant subject (Chemical Composition of Ambrosia trifida L. and Its Allelopathic Influence on Crops), Submitted to section: Plant Protection and Biotic Interactions,

https://www.mdpi.com/journal/plants/sections/plant_protection

Advances in Alternative Measures in Plant Protection”.

The experiment is sound, the manuscript is very interesting and well written, with an interesting set of well-presented results and adequate discussion. The text is clear and easy for readers and the appropriate topics are supported by the literature, the conclusions consistent with the evidence and arguments presented. The data are sufficient, so i recommend that the manuscript should be accepted after minor revision.

Specific comments

Please follow the comments in the Pdf version in the following lines

  • Line 28 please add these words (maize; sunflower) to keywords
  • Line 31 delete which
  • Lines 70 and 91 references needed
  • Lines 168 and 172 at 10 days
  • Line 170 at 14 days
  • Line 173 at 14 days
  • Line 187 at 7 days
  • Line 199 at 10 and 14 days (please add at before the number), do the same in all manuscript.
  • Line 265 new reference needed
  • Lines 273, 285 and 287 please add at before the number
  • Lines 327-342 please rewrite these sections according to your comment, your information and the rules of MDPI

Author Response

Response to Reviewer 3 Comments

Response: Dear Reviewer 3, we would like to thank you for your comments which have greatly contributed to the quality of our paper. It was rather difficult to correct the entire paper, since there were 3 Reviewers who asked for completely different corrections. We have done our best to accept all your suggestions and you can find the corrections in the attached paper highlighted in green. We sincerely hope that you will find our effort and the work we have put in correcting the paper useful and that our paper will now meet your criteria for being published in your journal.

Point 1: Line 28 please add these words (maize; sunflower) to keywords
Response: Corrected and highlighted in green.
Point 2: Line 31 delete which
Response:We have done the correction.
Point 3: Lines 70 and 91 references needed.
Response: Corrected and highlighted in green.
Point 4: Lines 168 and 172 at 10 days.
Response: Corrected and highlighted in green.
Point 5: Line 170 at 14 days.
Response: Corrected and highlighted in green.
Point 6: Line 173 at 14 days
Response: Corrected and highlighted in green.
Point 7: Line 187 at 7 days
Response: Corrected and highlighted in green.
Point 8: Line 199 at 10 and 14 days (please add at before the number), do the same in all manuscript.
Response: Corrected and highlighted in green.
Point 9: Line 265 new reference needed
Response: Corrected and highlighted in green.
Point 10: Lines 273, 285 and 287 please add at before number
Response: Corrected and highlighted in green.
Point 11: Lines 327-342 please rewrite these sections according to your comment, your information and the rules of MDPI
Response: Corrected and highlighted in green.

This manuscript is a resubmission of an earlier submission. The following is a list of the peer review reports and author responses from that submission.